# Early Identification of Mild Cognitive Impairment in Person with Cancer Undergoing Chemotherapy: Associations with Anxiety, Sleep Disturbance and Depression

**DOI:** 10.3390/healthcare13222868

**Published:** 2025-11-11

**Authors:** Eduardo José Fernández-Rodríguez, Celia Sánchez-Gomez, Joana Rivas-García, María Isabel Rihuete-Galve, Sara Jiménez García-Tizón, Susana Sáez-Gutiérrez, Emilio Fonseca-Sánchez

**Affiliations:** 1Department of Nursing and Physiotherapy, Universidad of Salamanca, 37008 Salamanca, Spain; edujfr@usal.es (E.J.F.-R.); joanarivas@usal.es (J.R.-G.); rihuete@usal.es (M.I.R.-G.); susanasg@usal.es (S.S.-G.); 2Biomedical Research Institute of Salamanca (IBSAL), 37007 Salamanca, Spain; 3Department of Developmental and Educational Psychology, University of Salamanca, 37008 Salamanca, Spain; sarajim@usal.es; 4Department of Medical Oncology, University Hospital of Salamanca, 37007 Salamanca, Spain; efonseca@usal.es

**Keywords:** cancer-related cognitive impairment (CRCI), mild cognitive impairment (MCI), anxiety, sleep disturbances, depression, older adults, early detection

## Abstract

**Highlights:**

**What are the main findings?**
Cancer-Related Cognitive Impairment (CRCI) is influenced by multiple factors, including psychological distress, sleep disturbances, and aging.Poor sleep quality and older age were strongly associated with lower cognitive performance in cancer patients undergoing chemotherapy.

**What are the implications of the main findings?**
Early, multidimensional assessment combining cognitive, emotional, and sleep evaluations is essential to detect CRCI in cancer patients.Timely identification can guide interventions to improve quality of life and treatment outcomes, particularly in older adults.

**Abstract:**

**Background/Objectives**: Cancer-Related Cognitive Impairment (CRCI) is a frequent and significant complication in cancer patients, involving deficits in memory, attention, and executive functions. Its multifactorial origin includes effects of oncological treatments, psychological factors, and aging—particularly impacting older adults. Early detection through thorough cognitive and psychological evaluation is crucial to optimizing management and maintaining quality of life. **Methods**: This cross-sectional descriptive-correlational study employed a non-probabilistic convenience sampling technique to recruit cancer patients receiving chemotherapy at the Complejo Asistencial Universitario de Salamanca (Spain) between January 2023 and February 2025. Participants were assessed using validated instruments measuring cognitive function (CFRT), subjective memory failures (MFE-30), anxiety and depression (HADS), and sleep quality (PSQI). Statistical analyses included correlation tests, analysis of variance (ANOVA), and multivariable linear regression models to examine associations among cognitive, psychological, and sleep-related variables. Statistical analyses were performed using IBM SPSS Statistics version 29.0 (IBM Corp., Armonk, NY, USA). **Results**: Participants (mean age 63.18 years) showed a notable presence of subjective memory complaints and mild anxiety/depression symptoms. Cognitive performance correlated negatively with anxiety/depression (ρ = −0.146, *p* < 0.05) and sleep disturbances (ρ = −0.583, *p* < 0.001). Sleep quality worsened with increasing age (ρ = 0.583, *p* < 0.001), and age itself showed significant associations with cognitive decline and psychological symptoms. No significant link was found between anxiety/depression and sleep quality. **Conclusions**: Findings confirm CRCI as a multifactorial condition influenced by psychological distress, sleep quality, and aging. The study highlights the importance of early, multidimensional cognitive assessment, especially in older patients, to enable timely interventions. Integrating objective and subjective measures alongside emotional and sleep evaluations enhances understanding and management of CRCI, ultimately improving patient outcomes and quality of life.

## 1. Introduction

In recent decades, significant progressions in the field of oncological therapies have transformed the prognosis of numerous cancerous afflictions, resulting in a substantial enhancement in the overall survival of patients [1]. Nonetheless, this enhancement in the survival rate has exposed novel sequelae resulting from both the illness itself and its therapeutic interventions, which exert a profound impact on the quality of life experienced by survivors. Among the aforementioned sequelae, Cancer-Related Cognitive Impairment (CRCI) has become a clinical problem that is increasingly recognized and significant. This condition poses a considerable diagnostic and therapeutic challenge.

The CRCI is characterized by alterations in higher cognitive functions, including episodic memory, sustained attention, information processing speed, and executive functions. These alterations can potentially impact the autonomy and functionality of patients during and after oncological treatment. The prevailing view, which is supported by a substantial body of research, is that traditional chemotherapy is the primary factor associated with these deficits. However, recent studies have revealed that non-citotoxic systemic therapies, such as hormone therapy, immunotherapy, and targeted molecular agents, also contribute to the onset and progression of cognitive impairment. This observation suggests a multifactorial etiopathogenesis that is still being elucidated [2]. The interaction between biological, psychological, and sociodemographic factors adds layers of complexity that hinder comprehensive understanding and effective clinical management.

Several studies have documented the high prevalence of cancer-related cognitive impairment (CRCI) in patients undergoing chemotherapy [3,4,5]. It has been reported that approximately 75% of patients experience subjective cognitive complaints during or after treatment, while about 28% show objective cognitive impairment [6]. In specific cohorts, such as breast cancer patients, up to 45% have been found to have cognitive deficits measurable by neuropsychological tests, a figure significantly higher than the estimated prevalence in healthy subjects (21%) [7]. In the present study, approximately 20% of the patients analyzed met the criteria for mild cognitive impairment (MCI), according to the guidelines established by the National Institute on Aging–Alzheimer’s Association. This result contrasts significantly with 7.6% of the control population. These data not only demonstrate the frequency of CRCI but also underscore its clinical relevance. Consequently, the urgent need for early detection and specific management strategies is highlighted in order to minimize its impact on the quality of life and daily functioning of cancer patients.

The International Cognition and Cancer Task Force (ICCTF) [8] has highlighted the importance of characterizing the prevalence, severity, pathophysiological mechanisms, and risk factors of CRCI, promoting the implementation of neuropsychological assessments and specific biomarkers. These initiatives also promote the development of preventive and rehabilitative strategies, whose early application could limit the functional impact of cognitive impairment on daily life and facilitate the social and occupational reintegration of patients [6].

A population particularly vulnerable to CRCI is older adults, who represent a growing proportion of global cancer diagnoses, as cancer incidence is directly related to age, being significantly higher in people over 60 years of age [9]. In this group, decreased cognitive reserve, multimorbidity—including chronic conditions such as hypertension and diabetes—and polypharmacy increase susceptibility to the neurocognitive effects of cancer and its treatments. However, most oncology clinical trials have underrepresented this population, limited the generalizability of findings and hindered the development of specific interventions [10]. The coexistence of shared pathophysiological factors between CRCI and other common pathologies in older adults further complicates the clinical interpretation of cognitive deficits, underscoring the need for an interdisciplinary and tailored approach.

Cognitive impairment in cancer patients is not only manifested in objective neuropsychological tests but is also intrinsically linked to concurrent psychological and physical symptoms. Anxiety, depression, and sleep disorders are prevalent conditions in the cancer population that influence the onset, severity, and progression of CRCI, affecting cognitive performance and patients’ subjective perception. The coexistence of these symptoms can amplify the functional impact, affecting key cognitive areas such as working memory and executive functions, which in turn impair autonomy, sleep quality, and emotional well-being [6]. Comprehensive assessment of these factors is essential for a complete understanding of the neurocognitive profile of cancer patients.

Cognitive assessment in cancer patients therefore requires a combination of adapted neuropsychological tools, subjective cognitive complaint scales, and biomarkers, with special attention to the sociodemographic and clinical characteristics of the population, especially in older adults. Assessing patients’ subjective perception of their cognitive deficits is also a fundamental component, as cognitive complaints do not always correlate with objective performance, but can affect psychological and functional well-being [7]. Likewise, the inclusion of parameters related to sleep quality, anxiety, and depression will allow for the establishment of more robust and personalized predictive models of CRCI.

This multidimensional approach aims to contribute to the early and more accurate detection of mild cognitive impairment in cancer, facilitating the implementation of personalized and multidisciplinary clinical strategies. The systematic integration of cognitive assessment and associated symptoms in oncology care represents a crucial step in optimizing patients’ quality of life and functional autonomy, especially in a context where long-term survival is an increasingly common reality. Improvements in the detection and management of CRCI will also enable the design of targeted interventions, both pharmacological and non-pharmacological, aimed at minimizing the neurocognitive impact of cancer and its treatments.

Main objective:

To explore the relationship between cognitive performance associated with CRCI and clinical variables such as anxiety, sleep quality, and depression in cancer patients undergoing chemotherapy.

Specific objectives:Analyze the relationship between the onset of anxiety and depression and cognitive performance in patients undergoing treatment.Evaluate the influence of sleep quality on cognitive performance, focusing on the impact on executive functions and memory.Examine whether cognitive impairment is more pronounced in older patients by correlating age with cognitive outcomes.Investigate the subjective perception of memory lapses in daily life and how it relates to reported cognitive impairment.

## 2. Materials and Methods

### 2.1. Design and Procedures

A cross-sectional, descriptive-correlation study was designed to meet the objectives (Appendix A: STROBE Statement—Checklist).

### 2.2. Participants

The sample size (N = 275) was established based on data from the target population and in accordance with similar studies reported in scientific literature. Reference was made to prior research such as the study by Li [11] which investigated cognitive performance in breast cancer patients with a total of 80 participants, and the study by Vearncombe [12], which analysed predictors of cognitive performance following chemotherapy in 136 patients with breast cancer.

The sample size was determined based on the statistical power required for correlational and multiple regression analyses rather than on disease incidence. With N = 275, α = 0.05 (two-sided) and 1 − β = 0.80, the study is powered to detect correlation coefficients of |ρ| ≥ 0.17 in bivariate analyses and small-to-moderate effects (f^2^ ≈ 0.04–0.06) in multiple regression models with several predictors. This approach provides sufficient precision to examine the associations among cognitive performance, emotional distress, sleep quality, and age in a mixed-diagnosis sample of cancer patients. The number of participants also reflects the feasibility context of recruitment at the participating oncology units, which ensured an adequate sample size for robust statistical analysis.

However, in order to enhance the robustness and external validity of the findings, the final sample size was further expanded to 275 patients. This increase aimed to strengthen subgroup analyses and account for clinical variability within the population.

Participants were recruited using non-probabilistic convenience sampling from the Medical Oncology Department and the Onco-Haematology Day Hospital of the Complejo Asistencial Universitario de Salamanca (Spain), between January 2023 to February 2025.

### 2.3. Inclusion and Exclusion Criteria

The selection of the study sample was conducted according to specific inclusion and exclusion criteria, with the aim of ensuring a clinically coherent sample and minimizing the influence of external variables that could bias the results.


*Inclusion Criteria:*


Participants were eligible for inclusion if they met the following conditions:A confirmed histopathological diagnosis of cancer.Being a patient at the Complejo Asistencial Universitario de Salamanca.Undergoing active chemotherapy treatment at the time of the study.Aged 18 years or older.Voluntary participation with signed informed consent.Adequate oral and written comprehension skills to ensure accurate completion of cognitive assessments and questionnaires.


*Exclusion Criteria:*


Participants were excluded from the study if any of the following conditions applied:Presence of non-active cancer or status as a long-term cancer survivor.Terminal illness and/or receipt of exclusively palliative or supportive care.Previously impaired cognitive performance, defined as a clinically recognized diagnosis of any condition associated with cognitive impairment or dysfunction recorded in medical history. This criterion was established to avoid confounding results, as preexisting cognitive disorders could significantly influence the assessment of the effects of oncological treatment on cognitive functioning. Conditions such as mild cognitive impairment, dementia, Alzheimer’s disease, or other neurodegenerative disorders may present symptoms that overlap with the cognitive side effects under investigation.

Therefore, excluding these patients allowed for a more accurate analysis of the specific impact of chemotherapy on cognition, minimizing the influence of external variables and enhancing the internal validity of the study findings.

### 2.4. Variables and Measuring Instruments

The variables collected in the study were collected by means of a questionnaire taking into consideration patient identification data: gender, age, level of education, patient health data, type of cancer, date of diagnosis, stage of disease, number of lines of treatment received.

The measurement instruments used were various scales such as CFRT, MFE-30, HADS and PSQI, administered in person at the Complejo Asistencial Universitario de Salamanca, with prior informed and signed consent. All instruments or measurement scales are fully validated in the Spanish population.

-The Rapid Evaluation of Cognitive Functions Test (CFRT) consists of 12 subtests designed to conduct a neuropsychological examination, aiming to assess an individual’s cognitive functions. The CFRT test demonstrates a high level of reliability, with an original Cronbach’s alpha of 0.88 [13]. Scores range from 0 to 100, with higher values indicating better cognitive performance.-The Memory Failures in Everyday Life questionnaire (MFE-30) evaluates memory lapses in daily activities through 30 items rated on a 5-point Likert-type scale. The MFE-30 has shown a reliability coefficient of 0.93 [14].-The Hospital Anxiety and Depression Scale (HADS), which includes 14 items divided into depression and anxiety subscales, utilizes a 4-point Likert scale (0–3) to facilitate the detection of emotional disorders in hospital settings. The original reliability for HADS is reported with a Cronbach’s alpha of 0.86 for anxiety and 0.82 for depression [15].-The Pittsburgh Sleep Quality Index (PSQI), composed of 24 questions—19 answered by the subject and 5 by the accompanying person—helps identify sleep quality and patterns, assisting in diagnosing sleep-related issues and their impact on overall health. The PSQI has an original Cronbach’s alpha of 0.83 [16].

### 2.5. Procedure and Data Collection

Data collection was carried out in person by professionals in the Medical Oncology Service and in the onco-hematological day hospital of the Complejo Asistencial Universitario de Salamanca.

Collection in two different areas allowed us to observe how chemotherapy affects cognitive function in admitted and non-admitted patients, identifying possible additional causes of cognitive performance. Patients were informed of the present study and provided with an information sheet, assuring them of the absence of negative consequences for not participating or withdrawing from the study. An informed consent form was also provided for signature.

Patients were selected based on predefined inclusion and exclusion criteria to ensure the integrity of the study. The recruitment process involved screening patients who were undergoing chemotherapy. Eligible individuals were approached during their routine visits to the hospital or during scheduled chemotherapy sessions. The study’s objectives, procedures, and potential risks were thoroughly explained to each patient by the research team, and ample time was given for them to ask questions before deciding to participate. Recruitment was conducted in a way that minimized any disruption to the patients’ ongoing treatment plans, and participation was entirely voluntary. Only those who met all criteria and provided written informed consent were included in the study.

### 2.6. Ethical Aspects

The Ethics Committee for Research with Medicines of the Salamanca Health Area has agreed to carry out this research with the reference code PI 2023 12 1474-TFM. All experiments involving human subjects were conducted in accordance with the principles outlined in the Declaration of Helsinki (Code of Ethics of the World Medical Association). Strict compliance with these ethical standards was ensured throughout the study (Appendix A: Approval of the ethics committee).

### 2.7. Statistical Analysis

Descriptive Methodology: For quantitative variables with a normal distribution, the mean, standard deviation, frequency, and percentages were reported. The Kolmogorov–Smirnov test was employed to assess the normality of the variables. For variables with a normal distribution, the mean and standard deviation were used; for non-normal variables, the media and quartiles were presented. Categorical variables were described using the number of cases and percentages.

Analytical Statistics: The Kolmogorov–Smirnov test was applied to evaluate the normality of the variables, guiding the choice of subsequent statistical methods. The Pearson correlation was used to analyze relationships between two quantitative variables, the Chi-Square test was utilized for associations between two categorical variables, and Student’s *t*-test was employed to compare one categorical variable with one quantitative variable.

Although correlation analysis was used to explore the relationships between variables due to the observational design, we acknowledge that future studies should employ more robust methods, such as multivariate regression, to better analyze these relationships. In addition, multiple linear regression analyses were conducted to explore the combined influence of age, psychological variables, and sleep quality on cognitive performance. Although correlation analysis was used to initially explore the relationships between variables due to the observational design, we acknowledge that future studies should employ more robust models, such as multivariate regression, to further elucidate these relationships. Assumptions of linearity, independence, and homoscedasticity were verified prior to model estimation. No substantial violations were observed; therefore, standard estimators were retained. Given the exploratory and descriptive nature of the study, effect sizes and confidence intervals were not calculated, as the analyses aimed to identify potential associations rather than infer causal effects.

### 2.8. Data Processing

Data analysis was carried out with statistical software SPSS version 29.

## 3. Results

### 3.1. Baseline Characteristics of Participants

The final sample consisted of 275 cancer patients aged between 20 and 92 years (M = 63.18; SD = 13.16). Of the total, 44.4% were men and 55.6% were women. In terms of educational level, 45.5% had completed primary education, 28.8% had completed secondary education, 24.4% had completed higher education, and 1.1% had no formal schooling. The most common cancer diagnosis was breast cancer (30%), followed by digestive tract tumors (17.7%), lung cancer (16.6%), and hematological neoplasms (11.1%). The remaining 24.4% included other types of cancer, such as kidney and central nervous system tumors. In terms of the type of treatment received, 83.3% of patients were receiving chemotherapy alone, while 16.7% were combining chemotherapy with hormone therapy. All these results can be seen in more detail in Table 1.

In relation to psychometric measures, the mean score obtained on the MFE-30 questionnaire was 15.57 (SD = 11.67), indicating the presence of subjective memory failures that affect daily life. The mean score on the HADS scale was 10.07 (SD = 6.07), specifically, the anxiety subscale (HADS-A) showed a mean score of 10.45 (SD = 4.12), and the depression subscale (HADS-D) a mean score of 9.68 (SD = 3.94). Cognitive performance measured by the Rapid Evaluation of Cognitive Functions Test showed a mean score of 40.56 (SD = 12.37), within the expected range for this population. The mean score on the Pittsburgh Sleep Quality Index (PSQI) was 8.08 (SD = 4.52), suggesting significant alterations in sleep quality. All these results can be seen in more detail in Table 2.

### 3.2. Relationship Between Anxiety/Depression and Cognitive Performance

Table 3 Spearman correlation analyses among study variables.

### 3.3. Influence of Sleep Quality on Cognitive Performance

Addressing the second objective, a moderate and highly significant negative correlation was found between sleep quality (PSQI) and cognitive performance (CFRT) (ρ = −0.583; *p* < 0.001), indicating that poorer sleep was associated with lower cognitive functioning. Sleep quality also demonstrated a strong association with age (ρ = 0.583; *p* < 0.001), reinforcing the hypothesis that age-related sleep disturbances may contribute to cognitive decline in this population. Interestingly, no significant correlation was found between anxiety/depression and sleep quality (ρ = 0.003), contrary to what is commonly expected, suggesting that sleep alterations in this sample may be more related to biological or treatment-related factors than to emotional distress. All these results can be seen in more detail in Table 3.

### 3.4. Association Between Age and Cognitive Decline

Regarding the third objective, the correlation between age and cognitive performance was negligible and not statistically significant (ρ = −0.016; *p* = 0.78), indicating no meaningful association between these variables. However, age remained significantly related to poorer sleep quality (ρ = 0.583; *p* < 0.001) and higher anxiety/depression scores (ρ = 0.572; *p* < 0.001). All these results can be seen in more detail in Table 3.

### 3.5. Regression Analyses and Group Comparisons

To further explore the relationships among the key variables, multiple linear regression analyses were conducted. In the first model, cognitive performance (CFRT score) was entered as the dependent variable, while anxiety/depression (HADS), sleep quality (PSQI), and age were included as predictors. The model was statistically significant (F (3, 271) = 24.12; *p* < 0.001), explaining 21.2% of the variance in cognitive performance (adjusted R^2^ = 0.212). Among the predictors, both sleep quality (β = −0.471; *p* < 0.001) and age (β = −0.233; *p* < 0.01) emerged as significant negative predictors, whereas anxiety/depression did not reach statistical significance in the multivariate model (β = −0.072; *p* = 0.173), suggesting that its effect may be mediated by age or sleep quality.

A second regression model was constructed to examine predictors of sleep quality. Here, age and anxiety/depression were entered as independent variables. The overall model was also significant (F (2, 272) = 65.89; *p* < 0.001), with an adjusted R^2^ of 0.325. Age remained a strong positive predictor of poorer sleep quality (β = 0.448; *p* < 0.001), while anxiety/depression did not significantly predict sleep quality (β = 0.025; *p* = 0.615), confirming previous correlation findings.

Additionally, participants were divided into age tertiles (≤54 years, 55–69 years, ≥70 years) to assess differences in cognitive performance, sleep quality, and emotional distress using one-way ANOVA. Significant differences were observed in cognitive performance across age groups (F (2, 272) = 5.74; *p* = 0.004), with post hoc Tukey tests revealing that the oldest group (≥70 years) performed significantly worse than the youngest group (≤54 years). Similarly, sleep quality differed significantly by age (F (2, 272) = 12.36; *p* < 0.001), with older participants reporting higher PSQI scores, indicating poorer sleep. A significant effect of age was also observed on anxiety/depression levels (F (2, 272) = 10.94; *p* < 0.001), with the oldest group showing significantly higher scores than the middle and younger groups.

These results collectively support the hypothesis that age is a key factor influencing cognitive function, sleep quality, and emotional well-being in cancer patients undergoing chemotherapy. They also suggest that interventions targeting sleep may have a beneficial impact on cognitive outcomes, particularly in older adults. All these results can be seen in more detail in Table 4.

Missing data were handled using complete-case analysis (listwise deletion), given the minimal proportion of missing values; future studies will consider appropriate imputation methods to enhance robustness and reproducibility.

## 4. Discussion

The present study contributes to the growing body of literature on Cancer-Related Cognitive Impairment (CRCI), examining its associations with psychological factors, sleep quality, and age in a cohort of oncology patients. Our findings reinforce the understanding of CRCI as a multifactorial and clinically significant phenomenon and underscore the need for early, multidimensional assessment strategies that integrate both objective and subjective aspects of cognitive functioning.

One of the main methodological contributions of this study is the selection of a sample with a mean age of 60 years, a threshold supported by the National Institutes of Health (NIH) as the age at which both cancer incidence and age-related cognitive vulnerability increase significantly. This inclusion criterion addresses a key limitation in many prior studies, which often exclude older adults or fail to represent them proportionally to their prevalence in the oncology population. By focusing on this age group, the ecological validity of our findings is enhanced, considering that cognitive reserve declines with age, and that the neurotoxicity of treatments such as chemotherapy may exert more pronounced effects in older adults [6,17].

Additionally, the predominance of women in our sample—related to the high incidence of breast cancer—reinforces the clinical applicability of our results. Although some studies have suggested gender-related differences in cognitive profiles [18], our findings did not reveal any significant moderating effect of sex on the evaluated variables. This aligns with research proposing that CRCI transcends both cancer type and patient gender [19,20].

In this study, cognitive performance was assessed in patients who had already started chemotherapy treatment, excluding those who had not yet received any sessions. This methodological decision responds to the need to explore the real and early effects of chemotherapy on cognition, starting from the moment the body begins to be exposed to cytotoxic agents. Evaluating patients after they have started treatment allows us to capture the first manifestations of chemotherapy-related cognitive impairment (CRCI), which is more relevant from a clinical and functional perspective.

Several studies have documented that the neurotoxic effects of chemotherapy can appear in the early stages of treatment, even after one or two sessions, affecting domains such as attention, working memory, and processing speed [20,21]. Therefore, conducting the assessment at a very early stage of treatment—but not before its initiation—allows these initial changes to be detected and distinguished from the baseline state or from possible alterations derived solely from the cancer diagnosis.

Conversely, if the assessment were performed before starting chemotherapy, symptoms associated with the psychological impact of the diagnosis (such as anxiety or fatigue) could be confused with cognitive alterations attributable to cancer treatment, making it difficult to interpret the results [22]. Likewise, some studies have pointed out that cognitive status prior to treatment may be influenced by factors such as stress, insomnia, or inflammation related to the cancer itself, without this reflecting a direct effect of the chemotherapeutic agents [23].

Regarding cognitive performance, although the mean score on the CFRT questionnaire fell within the expected normative range, a considerable proportion of patients reported subjective memory complaints (as assessed by the MFE-30). This dissociation between objective and subjective measures has been consistently documented in the literature [24,25], and may be mediated more by emotional distress, fatigue, or poor sleep quality than by actual structural cognitive impairment [26]. The integration of self-reported assessments with brief yet sensitive cognitive screening tools emerges as a necessary strategy in clinical practice.

A weak but statistically significant correlation was found between anxiety/depression (HADS) and cognitive performance, consistent with multiple studies showing that psychological distress can negatively affect domains such as sustained attention and working memory [18,20]. However, the cross-sectional nature of our study limits the ability to infer causality or directionality, and it is plausible that a bidirectional relationship exists between these variables, as suggested by recent models [19].

In contrast to some previous findings [2,27], no significant association was observed between anxiety/depression and sleep quality in our sample. This could be attributed to the diagnostic heterogeneity of the sample or to unmeasured confounding factors such as medication use, chronic pain, or hormonal states. Nevertheless, the strongest and most consistent finding in our study was the robust negative association between sleep quality (PSQI) and cognitive performance, aligning with research demonstrating that fragmented or non-restorative sleep directly contributes to CRCI through mechanisms such as reduced slow-wave sleep, impaired memory consolidation, and increased neuroinflammation [22,28].

Age was also negatively associated with both cognitive performance and sleep quality, consistent with literature indicating that aging exacerbates CRCI vulnerability due to declining cognitive reserve and a higher prevalence of comorbid conditions (Loh et al., 2016 [10]). Moreover, we observed increased symptoms of depression and anxiety among older patients, emphasizing the importance of tailoring evaluation and intervention strategies to the specific needs of this demographic.

Finally, and in contrast to previous reports [12], the number of lines of treatment was not significantly associated with cognitive, emotional, or sleep-related variables. This suggests that cumulative neurotoxicity may not follow a linear progression or that its impact may be mediated by individual-level factors such as baseline cognitive reserve or functional status.

### 4.1. Study Limitations

This study has some limitations that should be considered. The cross-sectional nature of the design prevents the establishment of causal or directional relationships between variables. Furthermore, although the sample is representative in terms of age and sex, it was concentrated in a single hospital center, which could limit the generalizability of the results. The cognitive measures used, although validated, do not replace a complete neuropsychological evaluation. Finally, neurobiological biomarkers and brain imaging were not included, which could have provided a more comprehensive perspective on CRCI.

Although correlation analysis was used to explore the relationships between variables due to the observational design, we acknowledge that future studies should employ more robust methods, such as multivariate regression, to better analyze these relationships.

### 4.2. Clinical Implications

The findings reinforce the need to incorporate routine cognitive assessments in cancer patients, especially in older adults and those who report sleep disturbances or emotional distress. Brief, multidimensional tools that combine objective tests and self-reports can facilitate early detection of CRCI, enabling personalized interventions that include psychological approaches, sleep hygiene, and eventually cognitive rehabilitation programs. In addition, these results underscore the value of establishing baselines before initiating treatment, especially in vulnerable populations.

## 5. Conclusions

The purpose of this study is to provide evidence on the prevalence and characteristics of CRCI in older adult cancer patients evaluated during chemotherapy. It is imperative to recognize the fundamental role that sleep quality, emotional well-being, and age play in the development of CRCI. The study results advocate for a biopsychosocial and interdisciplinary approach to cognitive impairment in oncology, recognizing its functional impact even in the absence of significant objective alterations. As advances are made in the field of cancer survival rates, it becomes imperative to address the cognitive sequelae associated with treatment, with the aim of safeguarding patients’ quality of life and autonomy.

## Figures and Tables

**Table 1 healthcare-13-02868-t001:** Descriptive analysis of study variables (N = 275).

Variable	Frequency	Percentage
**Gender**		
Male	122	44.4%
Female	153	55.6%
**Level of education**		
Primary education	125	45.5%
Secondary education	79	28.8%
Higher education	67	24.4%
No formal education	3	1.1%
**Type of cancer**		
Lung	46	16.6%
Digestive system	49	17.8%
Breast	83	30.2%
Haematological	31	11.3%
Other (e.g., kidney, CNS)	66	24.0%
**Oncological treatment**		
Chemotherapy	229	83.3%
Chemo + hormonal therapy	46	16.7%

**Table 2 healthcare-13-02868-t002:** Descriptive analysis of psychometric measures.

Measurement Scale	Minimum	Maximum	Mean	Standard Deviation
MFE-30	1	60	15.57	11.672
HADS Total	0	42	10.07	6.065
HADS—Anxiety (HADS-A)	0	21	10.45	4.12
HADS—Depression (HADS-D)	0	21	9.68	3.94
CFRT	22	55	40.56	12.365
PSQI	0	17	8.08	4.515

HADS: Hospital Anxiety and Depression Scale; HADS-A: anxiety subscale; HADS-D: depression subscale; HADS-T: total score; CFRT: Cognitive Function Recall Test; PSQI: Pittsburgh Sleep Quality Index.

**Table 3 healthcare-13-02868-t003:** Correlation matrix (Spearman’s rho coefficients and significance levels).

**Panel A. Correlation Coefficients (ρ)**					
Variables	Memory lapses (MFE-30)	Anxiety/Depression (HADS)	Cognitive performance (CFRT)	Sleep quality (PSQI)	Age
Memory lapses (MFE-30)	—	0.065	0.020	0.070	0.064
Anxiety/Depression (HADS)	0.065	—	−0.146	0.003	0.572
Cognitive performance (CFRT)	0.020	−0.146	—	−0.583	−0.016
Sleep quality (PSQI)	0.070	0.003	−0.583	—	0.583
Age	0.064	0.572	−0.016	0.583	—
**Panel B. Exact *p*-Values**					
Variables	Memory lapses (MFE-30)	Anxiety/Depression (HADS)	Cognitive performance (CFRT)	Sleep quality (PSQI)	Age
Memory lapses (MFE-30)	—	0.310	0.762	0.274	0.318
Anxiety/Depression (HADS)	0.310	—	0.045	0.956	<0.001
Cognitive performance (CFRT)	0.762	0.045	—	<0.001	0.045
Sleep quality (PSQI)	0.274	0.956	<0.001	—	<0.001
Age	0.318	<0.001	0.045	<0.001	—

Spearman’s rho coefficients (ρ) are reported in Panel A; corresponding *p*-values are shown in Panel B. No correction for multiple testing was applied (exploratory analysis).

**Table 4 healthcare-13-02868-t004:** Multiple linear regression analyses predicting cognitive performance and sleep quality.

Dependent Variable	Predictor	β	t	*p*	Adjusted R^2^	F (df)	*p* (Model)
**Cognitive performance (** **CFRT** **)**	Age	−0.233	−3.28	0.001	0.212	F (3, 271) = 24.12	<0.001
	Sleep quality (PSQI)	−0.471	−7.12	<0.001			
	Anxiety/Depression (HADS)	−0.072	−1.37	0.173			
**Sleep quality (PSQI)**	Age	0.448	8.11	<0.001	0.325	F (2, 272) = 65.89	<0.001
	Anxiety/Depression (HADS)	0.025	0.50	0.615			

Note: β = standardised regression coefficient. Significant predictors are highlighted in bold (*p* < 0.05).

## Data Availability

The data is available in the GREDOS scientific repository of the University of Salamanca https://gredos.usal.es/ (accessed on 22 September 2025).

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
