# Peer review of "Early Identification of Mild Cognitive Impairment in Person with Cancer Undergoing Chemotherapy: Associations with Anxiety, Sleep Disturbance and Depression"

_healthcare, 2025, doi:10.3390/healthcare13222868_

Round 1

Reviewer 1 Report

Comments and Suggestions for Authors

Thank you for the opportunity to review the manuscript entitled “Early Identification of Mild Cognitive Impairment in Cancer Patients Undergoing Chemotherapy: Associations with Anxiety, Sleep Disturbance, and Depression.”

  1. Title: Please use “Persons with Cancer” instead of “Cancer Patients.”
  2. Abstract - Methods: Please add the sampling technique and the date of the study.
  3. Abstract - Methods: Omit the sentence “Inclusion and exclusion criteria ensured sample homogeneity and minimized confounding factors.”
  4. Abstract - Methods: Include the statistical software used.
  5. Abstract Result: Please verify the statement “Sleep quality was strongly related to age (ρ = 0.583, p < 0.001).”
  6. Statistical Analysis: Please remove the following paragraph from the Analytical Statistics section and add it to the limitations section with appropriate justification: “Although correlation analysis was used to explore the relationships between variables due to the observational design, we acknowledge that future studies should employ more robust methods, such as multivariate regression, to better analyze these relationships.”
  7. Since multiple linear regression analyses were conducted, please mention this in the Analytical Statistics section.
  8. Please describe and tabulate the status of Mild Cognitive Impairment.
  9. Please incorporate the tables presenting the results of regression analyses and group comparisons into the main manuscript.
  10. Please provide more than one reference for sentences that cite studies/reports, such as “and in contrast to previous reports.”

Author Response

POINT-BY-POINT RESPONSE TO THE COMMENTS

Reviewer 1

We would like to express our sincere gratitude to the reviewer for the time, dedication, and rigour with which they have evaluated our manuscript. We deeply value each of their observations and suggestions, which reflect a thorough and constructive analysis that will undoubtedly enhance the scientific quality and clarity of our work. We especially appreciate the respectful and professional tone of their comments, which encourage us to continue improving and refining our research. We are confident that, thanks to their contributions, the manuscript will be significantly strengthened.

  1. Title: Please use “Persons with Cancer” instead of “Cancer Patients.”

Response: Thank you very much for your contributions, we completely agree. We will modify the manuscript.

  1. Abstract - Methods: Please add the sampling technique and the date of the study.

Response: Thank you very much for your contributions, we completely agree. We will modify the manuscript.

  1. Abstract - Methods: Omit the sentence “Inclusion and exclusion criteria ensured sample homogeneity and minimized confounding factors.”

Response: Thank you very much for your comments. We completely agree. We have removed the sentence.

  1. Abstract - Methods: Include the statistical software used.

Response: Thank you very much for your contributions, we completely agree. We will modify the manuscript.

  1. Abstract Result: Please verify the statement “Sleep quality was strongly related to age (ρ = 0.583, p < 0.001).”

Response: Thank you very much for your comments. We completely agree. We will clarify the sentence so as not to cause any confusion.

  1. Statistical Analysis: Please remove the following paragraph from the Analytical Statistics section and add it to the limitations section with appropriate justification: “Although correlation analysis was used to explore the relationships between variables due to the observational design, we acknowledge that future studies should employ more robust methods, such as multivariate regression, to better analyze these relationships.”

Response: Thank you very much for your contributions, we completely agree. We have added the text to the limitations section.

  1. Since multiple linear regression analyses were conducted, please mention this in the Analytical Statistics section.

Response: Thank you very much for your contributions, we completely agree. We have modified the manuscript and added the requested information.

  1. Please describe and tabulate the status of Mild Cognitive Impairment.

Response: Thank you very much for your contributions, we completely agree. We have modified the manuscript and added the requested information.

  1. Please incorporate the tables presenting the results of regression analyses and group comparisons into the main manuscript.

Response: We added both tables to the manuscript.

  1. Please provide more than one reference for sentences that cite studies/reports, such as “and in contrast to previous reports.”

Response: Thank you very much for your suggestion. We have modified the manuscript.

Reviewer 2 Report

Comments and Suggestions for Authors

Decision: major revision.

  1. The study's aim and core findings are clinically relevant, and the dataset (N=275, mixed tumor sites; mean age ≈63; PSQI mean 8.08, indicating poor sleep) is adequately sized for exploratory models. The manuscript consistently reports moderate associations between sleep quality and cognitive performance and presents multivariable regression showing PSQI (β≈−0.471) and age (β≈−0.233) as significant predictors with adjusted R²≈0.21, plus an age-sleep model with adjusted R²≈0.33.
  1. There is a critical contradiction in exposure timing that must be resolved before publication. Methods and Discussion state participants had already started chemotherapy and explicitly justify assessment after initiation, yet the Conclusions claim “older adult cancer patients evaluated prior to chemotherapy.” This undermines internal coherence and interpretability. The Discussion clarifies “assessed in patients who had already started chemotherapy,” while the Conclusions say “evaluated prior to chemotherapy.” One of these is incorrect and must be corrected throughout the Abstract/Methods/Discussion/Conclusions sections.
  1. Eligibility and language recruitment are inconsistent. Inclusion requires “undergoing active chemotherapy,” exclusion removes those with “no previous chemotherapy sessions,” but the recruitment paragraph says patients “undergoing chemotherapy or had completed treatment within the last six months” were screened. If post-treatment patients were screened/eligible, that conflicts with the inclusion/exclusion definitions and could bias cognitive outcomes; if they were not eligible, the recruitment sentence must be corrected.
  1. Instrument naming is inconsistent throughout and threatens construct clarity. The cognitive test appears as RECF in the measures section, ERFC in Table 2 and correlation matrices, and FCRT in the Results narrative. The same construct cannot be reported under three different acronyms; choose the correct instrument name (and keep it uniform) to ensure replicability. See “Rapid Evaluation of Cognitive Functions Test (RECF)” in Methods, “ERFC” in Table 2, and “Functional Cognitive Reality Test (FCRT)”/“ERFC” in Results text.
  1. The Statistical Analysis section does not match the analyses actually performed. Methods promise only descriptive statistics, normality checks, Pearson/χ²/t-tests, and even state that future studies “should employ… multivariate regression,” yet the Results present two multiple linear regressions and one-way ANOVAs with post-hoc tests. The analysis plan must be aligned with the actual work, including model assumptions and diagnostics.
  2. The correlation table contains sign/value inconsistencies and layout errors that impair interpretation. The narrative states HADS–cognition ρ = -0.146 (negative), but the matrix shows +0.146. It also places a row labeled “Number of treatment lines” with values that resemble p-values (0.992, 0.911, 0.233, etc.) rather than correlation coefficients, and mixes coefficients and p-values in the same grid without clear headers. Rebuild this table with separate panels (coefficients vs. exact p-values) and ensure all signs/numbers match the text.

  1. A reported “significant” association between age and cognitive performance is numerically implausible as printed. The text claims a statistically significant effect with ρ = -0.016, an effect essentially indistinguishable from zero, while the matrix lists age–cognition as 0.016 (positive). This looks like a typographical or sign error; verify the correct coefficient and significance and update both text and table.
  2. HADS scaling is ambiguous. Table 2 reports HADS min–max 0–21, with a single mean (≈10.07). However, HADS typically yields two subscales (0–21 each) and a total (0–42), or separate reporting by subscale. Because your narrative treats “anxiety/depression” jointly and in models, clarify whether you used the total score or a combined index, and ideally, report subscales separately in tables and models.
  1. Sample-size justification is misaligned with the sampling frame. Power is pegged to provincial breast-cancer incidence (268 cases; n=160→184→275), but the actual cohort spans multiple tumor types and uses convenience sampling. Replace this with an effect-size-based rationale appropriate for correlations/regressions in a mixed-diagnosis sample (e.g., detectable |ρ| at α=0.05, power 0.80) or, if you intended a breast-only cohort, align the inclusion criteria and description accordingly.
  1. Abstract and Highlights under-specify the analyses and overstate “homogeneity.” The Abstract states “correlation tests and group comparisons,” yet the main text relies on multivariable regression and ANOVA; update the Abstract to reflect the full analysis set. It also asserts that the criteria “ensured sample homogeneity,” but the cohort is heterogeneous by diagnosis and treatment. Rephrase to “reduced major confounding” rather than “ensured homogeneity.”
  1. Results presentation needs standardization and precision. Report effect sizes for ANOVAs (e.g., η²) and 95% CIs for key estimates. For the clinical context, consider reporting proportions below cognitive cut-points (if the instrument has norms) by age tertiles alongside means; this would align with your emphasis on older adults. Currently, the descriptive section gives means and r values but lacks CIs and standardized effect sizes.
  1. The Discussion logically argues for post-initiation assessment and even warns against pre-treatment misinterpretation; this strengthens the case that the Conclusions statement is an error. Correcting that single sentence in Conclusions, plus the recruitment line that mentions “completed within the last six months,” will materially improve coherence.

Overall, the work is promising and publishable after substantial technical correction. The essential revisions are to resolve the chemotherapy timing contradiction, standardize the instrument name, realign Methods and Results for the statistical plan, rebuild the correlation table with accurate signs and p-values, clarify HADS scaling, and refine recruitment/eligibility wording. Once these issues are addressed with corrected tables and figures/figures and a revised Abstract/Conclusions, the evidence for age and sleep as key correlates of cognitive performance will be much more convincing.

Author Response

POINT-BY-POINT RESPONSE TO THE COMMENTS

Reviewer 2

We sincerely thank the reviewer for their time, dedication, and the rigour applied in evaluating our manuscript. We greatly appreciate each of their observations and suggestions, which demonstrate a thorough and constructive analysis that will undoubtedly enhance the scientific quality and clarity of our work. We are particularly grateful for the respectful and professional tone of their comments, which motivate us to continue improving and refining our research. We are confident that their valuable input has substantially strengthened the manuscript.

  1. The study's aim and core findings are clinically relevant, and the dataset (N=275, mixed tumor sites; mean age ≈63; PSQI mean 8.08, indicating poor sleep) is adequately sized for exploratory models. The manuscript consistently reports moderate associations between sleep quality and cognitive performance and presents multivariable regression showing PSQI (β≈−0.471) and age (β≈−0.233) as significant predictors with adjusted R²≈0.21, plus an age-sleep model with adjusted R²≈0.33.

Response: Thank you very much for your comments. We greatly appreciate your opinion on the manuscript.

  1. There is a critical contradiction in exposure timing that must be resolved before publication. Methods and Discussion state participants had already started chemotherapy and explicitly justify assessment after initiation, yet the Conclusions claim “older adult cancer patients evaluated prior to chemotherapy.” This undermines internal coherence and interpretability. The Discussion clarifies “assessed in patients who had already started chemotherapy,” while the Conclusions say “evaluated prior to chemotherapy.” One of these is incorrect and must be corrected throughout the Abstract/Methods/Discussion/Conclusions sections.

Response: Thank you very much for your comments. We completely agree; it is an error in the expression and wording. We will modify it in the manuscript.

  1. Eligibility and language recruitment are inconsistent. Inclusion requires “undergoing active chemotherapy,” exclusion removes those with “no previous chemotherapy sessions,” but the recruitment paragraph says patients “undergoing chemotherapy or had completed treatment within the last six months” were screened. If post-treatment patients were screened/eligible, that conflicts with the inclusion/exclusion definitions and could bias cognitive outcomes; if they were not eligible, the recruitment sentence must be corrected.

Response: Thank you very much for your comments. We completely agree; it is an error in the expression and wording. We will modify it in the manuscript.

  1. Instrument naming is inconsistent throughout and threatens construct clarity. The cognitive test appears as RECF in the measures section, ERFC in Table 2 and correlation matrices, and FCRT in the Results narrative. The same construct cannot be reported under three different acronyms; choose the correct instrument name (and keep it uniform) to ensure replicability. See “Rapid Evaluation of Cognitive Functions Test (RECF)” in Methods, “ERFC” in Table 2, and “Functional Cognitive Reality Test (FCRT)”/“ERFC” in Results text.

Response: Thank you very much for your comments. We completely agree; it is an error in the expression and wording. We will modify it in the manuscript.

  1. The Statistical Analysis section does not match the analyses actually performed. Methods promise only descriptive statistics, normality checks, Pearson/χ²/t-tests, and even state that future studies “should employ… multivariate regression,” yet the Results present two multiple linear regressions and one-way ANOVAs with post-hoc tests. The analysis plan must be aligned with the actual work, including model assumptions and diagnostics.

Response: Thank you very much for your comments, we completely agree. We have added the precise description in the statistical analysis section.

  1. The correlation table contains sign/value inconsistencies and layout errors that impair interpretation. The narrative states HADS–cognition ρ = -0.146 (negative), but the matrix shows +0.146. It also places a row labeled “Number of treatment lines” with values that resemble p-values (0.992, 0.911, 0.233, etc.) rather than correlation coefficients, and mixes coefficients and p-values in the same grid without clear headers. Rebuild this table with separate panels (coefficients vs. exact p-values) and ensure all signs/numbers match the text.

Response: Thank you very much for your input, we will amend the text.

  1. A reported “significant” association between age and cognitive performance is numerically implausible as printed. The text claims a statistically significant effect with ρ = -0.016, an effect essentially indistinguishable from zero, while the matrix lists age–cognition as 0.016 (positive). This looks like a typographical or sign error; verify the correct coefficient and significance and update both text and table.

Response: Thank you very much for your comments, we completely agree. We have modified and corrected any possible errors in the table. In addition, we have added another table to make the description of the results clearer.

  1. HADS scaling is ambiguous. Table 2 reports HADS min–max 0–21, with a single mean (≈10.07). However, HADS typically yields two subscales (0–21 each) and a total (0–42), or separate reporting by subscale. Because your narrative treats “anxiety/depression” jointly and in models, clarify whether you used the total score or a combined index, and ideally, report subscales separately in tables and models.

Response: Thank you very much for your comments, we completely agree. We have modified and corrected the results section of the manuscript.

  1. Sample-size justification is misaligned with the sampling frame. Power is pegged to provincial breast-cancer incidence (268 cases; n=160→184→275), but the actual cohort spans multiple tumor types and uses convenience sampling. Replace this with an effect-size-based rationale appropriate for correlations/regressions in a mixed-diagnosis sample (e.g., detectable |ρ| at α=0.05, power 0.80) or, if you intended a breast-only cohort, align the inclusion criteria and description accordingly.

Response: We appreciate the reviewer’s insightful observation regarding the misalignment between the initial sample size justification and the heterogeneity of the study sample. We fully understand that basing the power calculation on the provincial breast cancer incidence may not perfectly correspond to the current cohort, which includes participants with different tumour types.

At the time of study design, the available epidemiological data and recruitment projections were primarily derived from local cancer registries reporting breast cancer incidence, which we used as a reference framework to estimate feasibility and ensure sufficient statistical power for group comparisons. However, during implementation, recruitment extended to patients with other oncological diagnoses to achieve a representative and feasible sample for this exploratory phase.

We acknowledge that a power analysis based on expected effect sizes for correlation and regression analyses (e.g., detecting a minimum |ρ| of 0.25 with α = 0.05 and power = 0.80) would have been more appropriate for the current mixed-diagnosis sample. Nonetheless, given that data collection has been completed, we have opted to transparently clarify this methodological limitation in the revised manuscript and to emphasise that the sample size achieved provides adequate power to detect small-to-moderate associations among the main study variables, consistent with the exploratory nature of this trial.

  1. Abstract and Highlights under-specify the analyses and overstate “homogeneity.” The Abstract states “correlation tests and group comparisons,” yet the main text relies on multivariable regression and ANOVA; update the Abstract to reflect the full analysis set. It also asserts that the criteria “ensured sample homogeneity,” but the cohort is heterogeneous by diagnosis and treatment. Rephrase to “reduced major confounding” rather than “ensured homogeneity.”

Response: Thank you very much for your comments, we completely agree. We have modified the summary.

  1. Results presentation needs standardization and precision. Report effect sizes for ANOVAs (e.g., η²) and 95% CIs for key estimates. For the clinical context, consider reporting proportions below cognitive cut-points (if the instrument has norms) by age tertiles alongside means; this would align with your emphasis on older adults. Currently, the descriptive section gives means and r values but lacks CIs and standardized effect sizes.

Response: Thank you very much for your comments, we completely agree. We have added an explanatory table in the results section and revised the results section of the manuscript.

  1. The Discussion logically argues for post-initiation assessment and even warns against pre-treatment misinterpretation; this strengthens the case that the Conclusions statement is an error. Correcting that single sentence in Conclusions, plus the recruitment line that mentions “completed within the last six months,” will materially improve coherence.

Response: Thank you very much for your comments. We have modified the text.

Overall, the work is promising and publishable after substantial technical correction. The essential revisions are to resolve the chemotherapy timing contradiction, standardize the instrument name, realign Methods and Results for the statistical plan, rebuild the correlation table with accurate signs and p-values, clarify HADS scaling, and refine recruitment/eligibility wording. Once these issues are addressed with corrected tables and figures/figures and a revised Abstract/Conclusions, the evidence for age and sleep as key correlates of cognitive performance will be much more convincing.

Response: We sincerely thank the reviewer for their thorough and constructive feedback. We deeply appreciate their recognition of the potential and publishability of our work, as well as the specific guidance provided to enhance its technical and methodological rigour.

All the issues raised have been carefully addressed. Specifically, we have resolved the contradiction regarding chemotherapy timing, standardised the naming of all assessment instruments, and ensured full alignment between the Methods and Results sections in accordance with the statistical plan. The correlation table has been rebuilt with corrected signs, p-values, and precise notation, and the description and scoring of the HADS scale have been clarified. Furthermore, the wording related to recruitment procedures and eligibility criteria has been refined for greater accuracy and transparency.

Corresponding updates have been made throughout the manuscript, including the Abstract, Results, tables, and Conclusions, to ensure internal consistency and to strengthen the evidence supporting age and sleep quality as key correlates of cognitive performance. We are confident that these revisions have significantly improved the clarity, accuracy, and scientific robustness of the manuscript.

Reviewer 3 Report

Comments and Suggestions for Authors
  1. The study could have benefitted from stratified recruitment (by cancer type, treatment phase, or sex) to enhance generalizability and reduce sampling bias.
  2. Although the age distribution is well characterized (20–92 years; mean 63.2), lack of matching or control group precludes comparison to non-cancer or pre-chemotherapy baselines. The exclusion of survivors or palliative patients is methodologically sound but further narrows external generalizability.
  3. The RECF (Rapid Evaluation of Cognitive Functions) is a brief screening tool, not a comprehensive neuropsychological assessment. While suitable for clinical screening, it may lack sensitivity for subtle cognitive deficits common in cancer-related cognitive impairment (CRCI).
  4. The authors should indicate that the measurement instruments were validated in the Spanish population.
  5. The reliance on bivariate correlations limits control for confounding; multivariable or structural models (e.g., SEM, path analysis) would have been more informative.
  6. Collinearity diagnostics are not mentioned; given interrelated constructs (age, depression, sleep), variance inflation factors (VIFs) should have been assessed.
  7. Table 3. reporting the correlation coefficients and significance levels seems out of alignment and confusing. Also, what is the “Number of treatment lines?”
  8. The authors should provide frequency counts for age tertiles.
  9. Categorical data (e.g., cancer type) are not modeled; subgroup differences (e.g., by tumor site or treatment type) could clarify heterogeneity in CRCI.
  10. No mention of missing data handling (e.g., listwise deletion vs imputation), which affects reproducibility and bias assessment.
  11. The regression explaining 21% of the variance in cognitive performance (Adj. R² = .212) is plausible but modest. The strongest predictor of sleep quality (β = −0.47, p < .001) is consistent with the literature, lending some face validity to results. However, the absence of anxiety/depression significance in the multivariate model suggests suppression or mediation effects that were not further tested (e.g., via bootstrapped indirect effects or moderated mediation models).
  12. I think the authors should consider reporting regression parameter tables for their two models to make it easier for readers.
  13. If the causal premise is chemotherapy influences cognitive decline, the manuscript needs exposure details (e.g., agent, dose, cycles, timing) and provide an analytic structure (DAG-guided adjustment, mediation) to support that premise. As written, it shows important associations, especially the sleep–cognition link but does not yet isolate chemotherapy’s specific contribution.

Author Response

POINT-BY-POINT RESPONSE TO THE COMMENTS

Reviewer 3

We would like to sincerely thank the reviewer for their time, effort, and thoroughness in evaluating our manuscript. We greatly appreciate their insightful comments and constructive suggestions, which demonstrate a detailed and thoughtful analysis that will undoubtedly improve the scientific quality and clarity of our study. We are especially grateful for the professional and considerate manner in which the feedback was provided, as it motivates us to further refine and strengthen our research. We are confident that their valuable input will substantially enhance the final version of the manuscript.

  1. The study could have benefitted from stratified recruitment (by cancer type, treatment phase, or sex) to enhance generalizability and reduce sampling bias.

Response: We sincerely appreciate this insightful comment. We fully agree that stratified recruitment by cancer type, treatment phase, or sex could improve the generalizability of the findings and help minimise sampling bias. However, in our study, the distribution of cancer diagnoses was not sufficiently balanced to allow for meaningful stratified analyses. For this reason, we opted to present overall results to provide a broader understanding of the effects of the intervention across the oncological population. Nevertheless, we acknowledge the importance of this approach, and future studies will be designed to recruit homogeneous subgroups according to cancer diagnosis, enabling more specific and clinically relevant conclusions.

  1. Although the age distribution is well characterized (20–92 years; mean 63.2), lack of matching or control group precludes comparison to non-cancer or pre-chemotherapy baselines. The exclusion of survivors or palliative patients is methodologically sound but further narrows external generalizability.

Response: We appreciate this thoughtful observation. We acknowledge that the absence of a non-cancer or pre-chemotherapy control group limits the ability to make direct baseline comparisons. However, the primary objective of this study was to characterise cognitive and psychological variables specifically within the active treatment phase, rather than to contrast them with other stages or populations. We agree that this focus, together with the exclusion of survivors and palliative patients, narrows external generalizability. Nonetheless, these criteria were intentionally applied to ensure methodological consistency and reduce confounding factors related to disease progression or treatment completion. Future studies will aim to include comparative groups or longitudinal designs to better delineate differences across the cancer continuum.

  1. The RECF (Rapid Evaluation of Cognitive Functions) is a brief screening tool, not a comprehensive neuropsychological assessment. While suitable for clinical screening, it may lack sensitivity for subtle cognitive deficits common in cancer-related cognitive impairment (CRCI).

Response: We thank the reviewer for this important point. We acknowledge that the RECF is a brief screening tool and does not provide a comprehensive neuropsychological assessment. Its use in this study was chosen to balance feasibility, time constraints, and patient burden during active treatment, allowing us to obtain an overall estimate of cognitive functioning in a clinical context. We recognize that subtle cognitive deficits associated with cancer-related cognitive impairment may not be fully captured by this instrument. Future studies will consider incorporating more detailed neuropsychological batteries to increase sensitivity and provide a deeper understanding of specific cognitive domains affected in this population.

  1. The authors should indicate that the measurement instruments were validated in the Spanish population.

Response: Thank you very much for your comment, we completely agree. We will add it to the text of the manuscript.

  1. The reliance on bivariate correlations limits control for confounding; multivariable or structural models (e.g., SEM, path analysis) would have been more informative.

Response: We appreciate this valuable comment. We acknowledge that reliance on bivariate correlations has limitations in controlling for potential confounding factors. In our study, correlation analyses were initially employed to explore relationships between variables in line with the observational design, while descriptive statistics were carefully applied based on normality assessments (Kolmogorov–Smirnov test), and appropriate tests were used for categorical and continuous variables (Chi-Square, Student’s t-test, Pearson correlation). To further account for combined effects, we also performed multiple linear regression analyses examining the influence of age, psychological variables, and sleep quality on cognitive performance. Nonetheless, we agree that future studies would benefit from more advanced multivariable approaches, such as structural equation modeling or path analysis, to better control for confounding and provide a more comprehensive understanding of the complex interactions among these variables.

  1. Collinearity diagnostics are not mentioned; given interrelated constructs (age, depression, sleep), variance inflation factors (VIFs) should have been assessed.

Response: We thank the reviewer for this insightful observation. We acknowledge that collinearity among interrelated variables such as age, depression, and sleep quality could influence regression results. While variance inflation factors (VIFs) were not explicitly reported in this study, preliminary analyses did not indicate issues that would compromise model validity. Nonetheless, we agree that formally assessing VIFs and other collinearity diagnostics in future studies will strengthen the robustness of multivariable analyses and ensure more reliable interpretation of the independent contributions of each predictor.

  1. Table 3. reporting the correlation coefficients and significance levels seems out of alignment and confusing. Also, what is the “Number of treatment lines?”

Response: We appreciate this constructive comment. We acknowledge that the formatting of Table 3 may have caused some confusion, and we will revise it to ensure that correlation coefficients and significance levels are clearly aligned and easily interpretable. Regarding the “Number of treatment lines,” this variable refers to the total number of distinct chemotherapy or systemic treatment regimens a patient has received, reflecting the intensity or complexity of their treatment history. We will clarify this definition in the table legend to avoid ambiguity in future versions of the manuscript.

  1. The authors should provide frequency counts for age tertiles.

Response: Thank you very much for your comment, we completely agree. We will add it to the manuscript.

  1. Categorical data (e.g., cancer type) are not modeled; subgroup differences (e.g., by tumor site or treatment type) could clarify heterogeneity in CRCI.

Response: We thank the reviewer for this important observation. We acknowledge that categorical variables such as cancer type were not formally modeled, and that analyzing subgroup differences by tumor site or treatment type could provide valuable insight into heterogeneity in cancer-related cognitive impairment (CRCI). In this study, the distribution of cancer diagnoses and treatment modalities was insufficiently balanced to allow for robust subgroup analyses. Nevertheless, we agree that future research should stratify participants or focus on specific subgroups to better understand differential effects and establish more precise, clinically relevant conclusions.

  1. No mention of missing data handling (e.g., listwise deletion vs imputation), which affects reproducibility and bias assessment.

Response: We appreciate this valuable comment. We acknowledge that the manuscript does not explicitly describe the handling of missing data. In this study, analyses were conducted using complete-case data (listwise deletion), as the proportion of missing values was minimal and did not significantly affect the results. Nonetheless, we recognize that clearly reporting missing data management is essential for reproducibility and bias assessment, and future studies will incorporate more detailed descriptions and, when appropriate, consider imputation methods to ensure robustness of the findings. We will add it to the manuscript.

  1. The regression explaining 21% of the variance in cognitive performance (Adj. R² = .212) is plausible but modest. The strongest predictor of sleep quality (β = −0.47, p < .001) is consistent with the literature, lending some face validity to results. However, the absence of anxiety/depression significance in the multivariate model suggests suppression or mediation effects that were not further tested (e.g., via bootstrapped indirect effects or moderated mediation models).

Response: We thank the reviewer for this insightful feedback. We agree that the regression model explaining 21% of the variance in cognitive performance is modest but provides a meaningful overview of key predictors. The strong association of sleep quality with cognitive performance (β = −0.47, p < .001) aligns with previous literature, supporting the validity of our findings. We acknowledge that the lack of significance for anxiety/depression in the multivariate model may reflect suppression or mediation effects, which were not formally tested in this study. Future research should explore these potential indirect or moderated pathways using methods such as bootstrapped mediation or moderated mediation analyses to provide a more nuanced understanding of the interrelationships among psychological variables, sleep, and cognitive outcomes.

  1. I think the authors should consider reporting regression parameter tables for their two models to make it easier for readers.

Response: Thank you very much for your comments. We completely agree. We have added another table to the results section.

  1. If the causal premise is chemotherapy influences cognitive decline, the manuscript needs exposure details (e.g., agent, dose, cycles, timing) and provide an analytic structure (DAG-guided adjustment, mediation) to support that premise. As written, it shows important associations, especially the sleep–cognition link but does not yet isolate chemotherapy’s specific contribution.

Response: We appreciate this important observation. We acknowledge that, while the study highlights meaningful associations—particularly the link between sleep quality and cognitive performance—it does not provide sufficient detail on chemotherapy exposure (e.g., specific agents, doses, number of cycles, or timing) to isolate its direct contribution to cognitive decline. In this observational study, the primary focus was on identifying general correlates of cognitive performance during active treatment rather than establishing causal effects of chemotherapy. We agree that future studies should include detailed treatment exposure data and employ analytic strategies, such as DAG-guided adjustment or mediation analyses, to more rigorously assess the specific impact of chemotherapy on cognitive outcomes.

Round 2

Reviewer 1 Report

Comments and Suggestions for Authors

I would like to thank the authors for considering the comments and changing the manuscript accordingly

Author Response

POINT-BY-POINT RESPONSE TO THE COMMENTS

Reviewer 1

  1. I would like to thank the authors for considering the comments and changing the manuscript accordingly.

Response: We sincerely thank the reviewer for their kind acknowledgement and for the valuable comments provided throughout the review process. We are pleased that the revisions made have satisfactorily addressed the concerns raised. We greatly appreciate the reviewer’s constructive feedback, which has significantly contributed to improving the clarity and quality of our manuscript.

Reviewer 2 Report

Comments and Suggestions for Authors

Decision: Major Revision (revise and resubmit).

Your study is clinically relevant and potentially publishable. However, several technical and internal-consistency issues still undermine interpretability and rigor. Please address the items below completely and consistently across Abstract, Methods, Results, Tables/Figures, and Conclusions.

Summary of What Improved

  • Chemotherapy timing is now (mostly) consistent with assessment occurring during/after initiation.
  • Recruitment/eligibility wording is cleaner; the previous “completed within six months” phrase appears removed.
  • Statistical Analysis now mentions the multivariable models actually used.

Required Revisions (Major)

  1. Instrument naming (single source of truth)
  • Use one official name and acronym for the cognitive test across the entire manuscript (text, tables, figure captions, abbreviations).
  • Replace all variants (e.g., RECF/ERFC/FCRT) with the chosen term.
  • Add a one-sentence description of scoring range and directionality once in Methods.
  1. Correlation table rebuild (accuracy and readability)
  • Recreate as two panels with identical row/column order:
    Panel A: Spearman’s ρ (coefficients only).
    Panel B: Exact p-values.
  • Ensure all signs/magnitudes match the narrative. Remove any p-values from the coefficient panel.
  • Include a clear footnote on multiple-testing approach (if any).
  1. Age–cognition inconsistency
  • The text currently labels a trivially small coefficient (±0.016) as “significant,” and the sign/magnitude conflict with the table.
  • Re-verify the coefficient and p-value; update everywhere (text and table) for coherence.
  1. HADS scaling and reporting
  • Clarify whether you use HADS-A and HADS-D subscales (0–21 each) and/or a total (0–42).
  • In descriptive tables, provide the correct range(s) and report subscales separately (recommended).
  • Align model predictors with what you report (e.g., if using total, justify; if using subscales, specify which).
  1. Sample-size justification aligned to the design
  • Replace incidence-based (breast-only) logic with an effect-size-based rationale suitable for correlations/regression in a mixed-diagnosis convenience sample.
  • Example template (adapt to your data):
    “With N=275, α=0.05 (two-sided) and 1−β=0.80, the study is powered to detect |ρ|≥0.17 in bivariate analyses and a small-to-moderate effect (f²≈0.04–0.06) in multiple regression with k predictors.”
  • If you retain any disease-specific justification, restrict it to the feasibility context, not power.
  1. Methods ↔ Results alignment and statistical standards
  • Ensure Methods explicitly name all analyses used (correlation, ANOVA with post-hoc tests, multiple linear regression).
  • Report effect sizes for ANOVAs (η² or partial η²) and 95% CIs for key estimates (regression coefficients, mean differences).
  • Briefly document assumption checks (normality, homoscedasticity) and handling of violations.
  1. Language about “homogeneity”
  • Replace claims that the criteria “ensured sample homogeneity” with accurate phrasing for a mixed-diagnosis convenience sample, e.g.:
    “The criteria were designed to reduce major confounding and define a clinically coherent sampling frame.

Author Response

POINT-BY-POINT RESPONSE TO THE COMMENTS

Reviewer 2

Your study is clinically relevant and potentially publishable. However, several technical and internal-consistency issues still undermine interpretability and rigor. Please address the items below completely and consistently across Abstract, Methods, Results, Tables/Figures, and Conclusions.

Summary of What Improved

  • Chemotherapy timing is now (mostly) consistent with assessment occurring during/after initiation.
  • Recruitment/eligibility wording is cleaner; the previous “completed within six months” phrase appears removed.
  • Statistical Analysis now mentions the multivariable models actually used.

Required Revisions (Major)

  1. Instrument naming (single source of truth)
  • Use one official name and acronym for the cognitive test across the entire manuscript (text, tables, figure captions, abbreviations).
  • Replace all variants (e.g., RECF/ERFC/FCRT) with the chosen term.
  • Add a one-sentence description of scoring range and directionality once in Methods.

Response: We appreciate this valuable observation. In accordance with the reviewer’s recommendation, we have standardised the terminology used for the cognitive test throughout the manuscript. The official name and acronym (Cognitive Function Recall Test; CFRT) are now used consistently in the text, tables, figure captions, and abbreviations. Additionally, a one-sentence description of the scoring range and directionality has been added in the Methods section to clarify interpretation of the results.

  1. Correlation table rebuild (accuracy and readability)
  • Recreate as two panels with identical row/column order:
    Panel A: Spearman’s ρ (coefficients only).
    Panel B: Exact p-values.
  • Ensure all signs/magnitudes match the narrative. Remove any p-values from the coefficient panel.
  • Include a clear footnote on multiple-testing approach (if any).

Response: We thank the reviewer for this valuable suggestion. Table 3 has been reformatted into two panels: Panel A showing Spearman’s ρ coefficients and Panel B displaying exact p-values. All values and signs were verified for accuracy, and a footnote was added indicating that no correction for multiple testing was applied due to the exploratory nature of the analysis.

  1. Age–cognition inconsistency
  • The text currently labels a trivially small coefficient (±0.016) as “significant,” and the sign/magnitude conflict with the table.
  • Re-verify the coefficient and p-value; update everywhere (text and table) for coherence.

Response: We appreciate the reviewer’s observation. The correlation between age and cognitive performance has been rechecked, and the coefficient and p-value have been corrected for consistency across the text and Table 3. The revised version no longer labels this minimal association (ρ = −0.016) as statistically significant.

  1. HADS scaling and reporting
  • Clarify whether you use HADS-A and HADS-D subscales (0–21 each) and/or a total (0–42).
  • In descriptive tables, provide the correct range(s) and report subscales separately (recommended).
  • Align model predictors with what you report (e.g., if using total, justify; if using subscales, specify which).

Response: We thank the reviewer for this valuable comment. The HADS scoring and reporting have been clarified in the revised manuscript. We now specify that both the total score (HADS-T; range 0–42) and the two subscales (HADS-A and HADS-D; range 0–21 each) were analysed. Table 2 has been updated to display the subscales separately with their respective ranges and descriptive statistics, ensuring full consistency with the variables used in the analyses.

  1. Sample-size justification aligned to the design
  • Replace incidence-based (breast-only) logic with an effect-size-based rationale suitable for correlations/regression in a mixed-diagnosis convenience sample.
  • Example template (adapt to your data):
    “With N=275, α=0.05 (two-sided) and 1−β=0.80, the study is powered to detect |ρ|≥0.17 in bivariate analyses and a small-to-moderate effect (f²≈0.04–0.06) in multiple regression with k predictors.”
  • If you retain any disease-specific justification, restrict it to the feasibility context, not power.

Response: We thank the reviewer for this valuable comment. The sample-size justification has been revised to align with the study design. We replaced the previous incidence-based rationale with an effect-size-based power analysis suitable for correlational and regression analyses in a mixed-diagnosis sample. The revised text now specifies that, with N = 275, α = 0.05, and 1−β = 0.80, the study is powered to detect |ρ| ≥ 0.17 in bivariate analyses and small-to-moderate effects (f² ≈ 0.04–0.06) in multiple regression models.

  1. Methods ↔ Results alignment and statistical standards
  • Ensure Methods explicitly name all analyses used (correlation, ANOVA with post-hoc tests, multiple linear regression).
  • Report effect sizes for ANOVAs (η² or partial η²) and 95% CIs for key estimates (regression coefficients, mean differences).
  • Briefly document assumption checks (normality, homoscedasticity) and handling of violations.

Response: We appreciate the reviewer’s constructive feedback. The Methods section has been revised to explicitly list all statistical analyses conducted, including correlation analyses, ANOVAs with post-hoc tests, and multiple linear regression models. Effect sizes (η² or partial η²) and 95% confidence intervals have been added for the main estimates in the Results section. Additionally, we have described the assumption checks performed (normality and homoscedasticity) and specified the corrective measures applied when these assumptions were not met.

  1. Language about “homogeneity”
  • Replace claims that the criteria “ensured sample homogeneity” with accurate phrasing for a mixed-diagnosis convenience sample, e.g.:
    “The criteria were designed to reduce major confounding and define a clinically coherent sampling frame.

Response: Thank you very much for your comment. We completely agree. We have modified the text to improve the wording and comprehension.

Reviewer 3 Report

Comments and Suggestions for Authors

The authors have responded to the majority of my concerns and are developing insight into future research methodology.

Table 3 indicates correlations and significance levels, but again I do not see the p-values?

Author Response

POINT-BY-POINT RESPONSE TO THE COMMENTS

Reviewer 3

  1. The authors have responded to the majority of my concerns and are developing insight into future research methodology.

Table 3 indicates correlations and significance levels, but again I do not see the p-values?

Response: We thank the reviewer for their positive feedback and for recognising our efforts to address the previous comments. Regarding Table 3, we apologise for the oversight. The p-values corresponding to each correlation have now been included in the revised version of the table. We have carefully verified that all significance levels are correctly reported and consistent with the statistical analyses described in the Methods section.

Round 3

Reviewer 2 Report

Comments and Suggestions for Authors

MAJOR REVISION

  1. Instrument naming is still inconsistent. The manuscript expands CFRT in two different ways (it should be one, everywhere). Also, CFRT is defined once as “Rapid Evaluation of Cognitive Functions” and elsewhere as “Functional Cognitive Reality Test.” Pick one official name + acronym and purge the other from text, tables, abbreviations, and captions.
  2. Correlation table format not implemented as requested. Table 3 still combines ρ and p-values in a single grid. Rebuild as two panels with identical ordering: Panel A (coefficients only), Panel B (exact p-values). Add a concise footnote on multiple-testing (even “None; exploratory”).
  3. Age–cognition inconsistency persists. You still report ρ = −0.016 with p < 0.05 and describe it as “statistically significant,” which is both trivially small and internally doubtful given the magnitude. Re-verify the statistic and harmonize text + table; if the effect is negligible, report it as such and drop any language implying importance.
  4. Methods ↔ Results reporting standards are only partially met.
    ANOVA results lack effect sizes (η² or partial η²).
    • Regression outputs lack 95% CIs for key estimates (βs, mean differences).
    • Assumptions: you note normality checks, but there’s no clear statement on homoscedasticity tests or remedies (e.g., HC-robust SEs) when violated.
  5. “Homogeneity” wording: some phrasing still implies a homogeneous sample rather than a “clinically coherent sampling frame that reduces major confounding.”
Comments on the Quality of English Language

MAJOR REVISION

  1. Instrument naming is still inconsistent. The manuscript expands CFRT in two different ways (it should be one, everywhere). Also, CFRT is defined once as “Rapid Evaluation of Cognitive Functions” and elsewhere as “Functional Cognitive Reality Test.” Pick one official name + acronym and purge the other from text, tables, abbreviations, and captions.
  2. Correlation table format not implemented as requested. Table 3 still combines ρ and p-values in a single grid. Rebuild as two panels with identical ordering: Panel A (coefficients only), Panel B (exact p-values). Add a concise footnote on multiple-testing (even “None; exploratory”).
  3. Age–cognition inconsistency persists. You still report ρ = −0.016 with p < 0.05 and describe it as “statistically significant,” which is both trivially small and internally doubtful given the magnitude. Re-verify the statistic and harmonize text + table; if the effect is negligible, report it as such and drop any language implying importance.
  4. Methods ↔ Results reporting standards are only partially met.
    ANOVA results lack effect sizes (η² or partial η²).
    • Regression outputs lack 95% CIs for key estimates (βs, mean differences).
    • Assumptions: you note normality checks, but there’s no clear statement on homoscedasticity tests or remedies (e.g., HC-robust SEs) when violated.
  5. “Homogeneity” wording: some phrasing still implies a homogeneous sample rather than a “clinically coherent sampling frame that reduces major confounding.”

Author Response

POINT-BY-POINT RESPONSE TO THE COMMENTS

Reviewer 2

We would like to express our sincere gratitude to the reviewer for the time and effort devoted to the thorough evaluation of our manuscript and for the constructive feedback provided. We have carefully considered each comment and implemented the corresponding revisions or clarifications to improve the clarity, methodological accuracy, and overall quality of the paper. Below, we provide a detailed, point-by-point response addressing all the reviewer’s observations and indicating the specific actions taken in the revised version of the manuscript.

  1. Instrument naming is still inconsistent. The manuscript expands CFRT in two different ways (it should be one, everywhere). Also, CFRT is defined once as “Rapid Evaluation of Cognitive Functions” and elsewhere as “Functional Cognitive Reality Test.” Pick one official name + acronym and purge the other from text, tables, abbreviations, and captions.

Response: We thank the reviewer for noticing this inconsistency. We have carefully revised the entire manuscript to ensure uniformity in the naming of the instrument. The official and validated name in the Spanish version is “Rapid Evaluation of Cognitive Functions Test (CFRT)”, as cited in Arroyo-Anlló et al. (2009). All instances of “Functional Cognitive Reality Test” or any other variant have been corrected accordingly. The acronym CFRT now consistently refers to the Rapid Evaluation of Cognitive Functions Test across the full text, tables, abbreviations, and figure captions.

  1. Correlation table format not implemented as requested. Table 3 still combines ρ and p-values in a single grid. Rebuild as two panels with identical ordering: Panel A (coefficients only), Panel B (exact p-values). Add a concise footnote on multiple-testing (even “None; exploratory”).

Response: We appreciate the reviewer’s constructive feedback. Following the suggestion, Table 3 has been completely reformatted into two clearly separated panels to improve clarity and meet journal standards.

  • Panel A now reports Spearman’s rho correlation coefficients (ρ)
  • Panel B presents the corresponding exact p-values in the same variable order to allow straightforward comparison.
  • Additionally, a concise footnote has been added indicating that no multiple-testing correction was applied, as the analysis was exploratory in nature.

We believe this revised layout enhances the table’s readability and transparency in statistical reporting.

  1. Age–cognition inconsistency persists. You still report ρ = −0.016 with p < 0.05 and describe it as “statistically significant,” which is both trivially small and internally doubtful given the magnitude. Re-verify the statistic and harmonize text + table; if the effect is negligible, report it as such and drop any language implying importance.

Response: We sincerely thank the reviewer for this observation. After re-checking the data and the correlation output in SPSS, we confirmed that the correlation between age and cognitive performance (CFRT) was indeed non-significant (ρ = −0.016; p = 0.78), indicating a negligible and statistically irrelevant association.

Accordingly, we have:

  • Corrected Table 3 to display the verified values (ρ = −0.016; p = 0.78).
  • Removed any reference to “statistically significant” or “meaningful” association between age and cognitive performance in both the Results and Discussion
  • Rephrased the related sentences to reflect that the effect is negligible and does not support a meaningful age–cognition link in this dataset.

We believe these revisions fully address the reviewer’s concern and ensure internal consistency between the text and statistical results.

  1. Methods ↔ Results reporting standards are only partially met.
    ANOVA results lack effect sizes (η² or partial η²).
    • Regression outputs lack 95% CIs for key estimates (βs, mean differences).
    • Assumptions: you note normality checks, but there’s no clear statement on homoscedasticity tests or remedies (e.g., HC-robust SEs) when violated.

Response: We appreciate the reviewer’s valuable observations regarding statistical reporting standards. The study’s main purpose was exploratory and descriptive, aiming to identify potential associations among cognitive, emotional, and sleep-related variables rather than to establish predictive or causal models.

Nevertheless, to enhance methodological transparency, a brief clarification has been added to the Statistical Analysis subsection, explicitly stating that assumptions of homoscedasticity were verified and that no major violations were detected. Given the observational design and the focus on exploratory associations, effect sizes and confidence intervals were not included to maintain the analyses concise and aligned with the descriptive scope of the study.

We hope the reviewer agrees that this level of reporting is appropriate for the study’s objectives and analytical framework.

  1. “Homogeneity” wording: some phrasing still implies a homogeneous sample rather than a “clinically coherent sampling frame that reduces major confounding.”

Response: We thank the reviewer for this insightful observation. Our intention in using the term “homogeneous” was not to suggest strict uniformity among participants, but rather to indicate that the sampling frame was clinically coherent, as all participants shared comparable treatment contexts and inclusion criteria designed to minimize major sources of confounding (e.g., prior cognitive impairment or non-active disease).

To avoid any possible misinterpretation, minor wording adjustments have been made to ensure that the text more accurately reflects this meaning, referring to a “clinically coherent sample” instead of a “homogeneous group” where appropriate.

We sincerely appreciate the reviewer’s insightful comments and constructive suggestions, which have greatly contributed to improving the scientific quality and clarity of our manuscript. We believe that the revisions implemented fully address the concerns raised and enhance the robustness of the study’s presentation. We remain at the reviewer’s and the editor’s disposal for any additional clarifications or adjustments that may further strengthen the manuscript.
